# Reinvestigating the Preferential Enrichment of DL-Arginine Fumarate: New Thoughts on the Mechanism of This Far from Equilibrium Crystallization Phenomenon

**DOI:** 10.3390/molecules27248652

**Published:** 2022-12-07

**Authors:** Clément De Saint Jores, Clément Brandel, Marie Vaccaro, Najla Gharbi, Isabelle Schmitz-Afonso, Pascal Cardinael, Rui Tamura, Gérard Coquerel

**Affiliations:** 1Institut de Chimie Organique et Analytique, University of Orléans, CNRS UMR 7311, CEDEX 2, 45067 Orléans, France; 2Rouen Normandie, FR3038, SMS, UR 3233, 76000 Rouen, France; 3University Rouen Normandie, SMS, UR 3233, 76000 Rouen, France; 4Normandie Univ, COBRA UMR 6014 et FR 3038 Univ Rouen, INSA Rouen, CNRS IRCOF, 1 Rue Tesnière, 76821 Mont-Saint-Aignan Cedex, France; 5Graduate School of Human and Environmental Studies, Kyoto University, Kyoto 606-8501, Japan

**Keywords:** crystallization, chiral resolution, symmetry breaking, preferential enrichment

## Abstract

Preferential enrichment (PE) is a crystallization process, starting from either a racemic of slightly enantio-enriched solution (ca. +5%) that results in a high enantiomeric excess in the liquid phase (>+90%ee) and a slight opposite excess in the deposited crystals (−2 to −5%ee). The mechanism(s) of this symmetry-breaking phenomenon is (are) still a matter of debate since it eludes rationalization by phase diagram formalism. In this publication, we thoroughly reinvestigate the PE phenomenon of arginine fumarate by using a new approach: the process is monitored by introducing isotopically labeled arginine enantiomers into the crystallization medium to better understand the mass exchanges during crystallization. These experiments are supported by chiral HPLC-MS/MS. This study permits re-evaluating the criteria that were thought mandatory to perform PE. In particular, we show that PE occurs by a continuous exchange between the solution and the crystals and does not require the occurrence of a solvent-mediated solid–solid phase transition.

## 1. Introduction

The production of enantiopure material is of paramount importance to several industries, ranging from fine chemicals and pharmaceutics [1] to optics and photonics [2]. When two chiral molecules are mirror related, they are called enantiomers. Two enantiomers have identical scalar properties but opposite vectorial properties and may behave differently in asymmetric media, such as living organisms [3]. For instance, the counter enantiomer of a chiral drug may have reduced, different or even adverse pharmacological properties compared to the active enantiomer. Two opposite enantiomers also have the same enthalpy of formation: starting from an achiral precursor, both enantiomers will be produced in equal amounts, thus giving rise to a racemic mixture. Asymmetric synthesis can be used to establish stereoselective access to the desired enantiomer but requires the use of expensive catalyzers. This is why the separation of enantiomers from their racemic mixture, i.e., chiral resolution, is most often preferred [4].

Beside preparative chiral chromatography [5], resolution by crystallization has been used for more than decades to separate enantiomers. Nowadays, Pasteurian resolution [6] preferential crystallization [7] and deracemization [8] are well documented and some of these techniques are even used at the industrial scale. Most of these techniques require that the mixture of enantiomers crystallizes as a physical mixture of homochiral particles (i.e., a conglomerate) rather than a new phase built with both enantiomers (i.e., a racemic compound or a solid solution). Yet, it is estimated that ca. 95% of enantiomeric mixtures crystallize as racemic compounds and only ca. 5% as conglomerates (solid solutions being an even rarer occurrence) [9]. To circumvent this issue, a screening of conglomerate is usually performed by either slight chemical modification of the molecule, salt or solvate formation and in rarer occurrences, co-crystal preparation [10]. For a molecule with an acid–base character, salt screening remains the preferred route [11]. Once the candidate substance is identified, the crystallization behavior of the mixture of enantiomer in the selected solvent is best rationalized with the use of a phase diagram [12].

The preferential enrichment (PE hereafter) phenomenon was discovered in 1993 by the team of Prof. Tamura at Kyoto University, initially in a family of organic salts (first generation of compounds) [13]. A second generation of compounds (consisting of amino acids [14] and other chiral pharmaceutical drugs [15]) has since been discovered and the results obtained ever since have been reviewed in a recent article [16]. By recrystallization of a racemic solid under considerably high supersaturation (usually, β > 6) in stagnant conditions, PE results in a strong enrichment of the mother liquor (>+90%ee) in one enantiomer and in a slight enrichment (−2% to −5%ee) of the deposited crystal in the opposite enantiomer. The red line in Figure 1a corresponds to this equilibrium. Such behavior is regarded as a spontaneous chiral symmetry-breaking phenomenon: according to the rules of phase equilibria. Starting from a purely racemic mixture, both the mother liquor and the deposited solid should remain racemic by following the blue tie lines in Figure 1a: PE is thus a crystallization process with violation of stable and metastable equilibria [17].

The mechanism by which this startling phenomenon occurs has been a matter of debate since its discovery. There is strong evidence that PE proceeds via the convergence of a number of circumstances: (i) First, at the solvated state homochiral associations are highly favored over heterochiral assemblies, most likely as 1D chains or 2D sheets [18]. (ii) Under considerable supersaturation conditions, a first metastable phase crystallizes using these homochiral pre-associations as building blocks regardless of their chirality [19], thus resulting in a large number of stacking faults and in zones with enantiopure content (Figure 1b). (iii) This metastable solid solution undergoes a redissolution of its enantiopure zones, a phenomenon favored by the high solubility of the pure enantiomer [20,21]. The first generation of compounds exhibited solvent-mediated polymorphic transitions concomitantly with such redissolution events. No such transition was clearly evidenced for the second generation of compounds. (iv) As the crystals redissolve the enantiomer in excess, the solution reaches high enantiomeric content and the deposited crystals, a slight opposite excess. This implies that the produced crystal structure must, to a certain extent, accommodate each enantiomer in the crystallographic sites (i.e., a solid-solution-type behavior) [22]. Therefore, the mechanism schematized in Figure 1b implies that a number of criteria must be met for the occurrence of PE: (a) The solubility of the pure enantiomer should be much higher than that of the racemic mixture. This criterion is actually regarded as one of the most important conditions. (b) Pre-nucleation aggregates should consist of homochiral (either 1D or 2D) assemblies. (c) The racemic form should have a solid-solution-like behavior allowing each enantiomer to occupy the same molecular sites, at least to a certain extent. In a recent paper, we demonstrated the importance of this criterion in the case of Tryptophan Ethyl Ester Hydrochloride, a molecular salt that satisfies criterion (a) but which cannot show PE due to the absence of solid solution behavior even under high β (β is defined as the ratio concentration by solubility) [23]. (d) The occurrence of polymorphic transition during crystallization has also been regarded as a prerequisite although no such event has been spotted for second-generation compounds.

The 1:1 compound between DL-arginine and fumaric acid (Figure 1, ArgFum hereafter) is one representative of the second generation of compounds and exhibits a highly efficient PE phenomenon. The crystal structure of DL-ArgFum has been solved as a racemic compound in the P-1 space group and the crystal structure has been extensively described in [24]. However, no other polymorph of this substance has been reported to date. In this structure, arginine forms heterochiral dimers via head-to-tail strong hydrogen bonds between the carboxylate and the guanidyl groups. These dimers are arranged in 2D layers which are stacked along **c** and are bridged via layers of fumaric acid molecules. The pure enantiomer of this molecular salt can be obtained by dehydration of the hydrated form and although its crystal structure is unknown, its powder diffraction pattern is different from that of the racemic form, thus indicating a different structure. The ‘anticonglomerate’ behavior of this system is confirmed by the large solubility differences between the racemic form (poorly soluble) and the pure enantiomer (very soluble) in water. Thus, like most representatives of the second generation [24], the DL-ArgFum system satisfies several of the above criteria, including the presence of homophile associations [25] but the absence of polymorphic behavior and the fact that no solvent-mediated solid transition has been observed [16], questions the mandatory nature of criterion (d). More recently, Manoj et al. [26] reported the crystal structure of two new salts in this system: a (1:1:1) monohydrated salt of racemic arginine and fumaric acid and a (1:2:1) monohydrated salt of L-arginine and fumaric acid.

Using chiral HPLC-UV and HPLC-MS/MS methods, the aim of the present article is to investigate further the mechanism of PE of DL-ArgFum by careful mass balance assessment and systematic monitoring of the chiral composition of each phase. By shedding light on the non-classical relationship existing between the liquor and the solid phase during this process, we intend to establish an updated version of the mechanism of PE.

## 2. Results

### 2.1. Performance of the Standard Protocol for the PE of Arginine Fumarate

In this manuscript, all enantiomeric excesses (ee) of L-ArgFum are positive (+), and excesses of D-ArgFum are negative (−). In order to establish a reference situation concerning the preferential enrichment (PE) of arginine fumarate (ArgFum), the standard protocol (see Materials and Methods section) was performed approximately 50 times. The process starts with a slight ee of (+) 5%ee (red cross in Figure 2). The typical results of these experiments are detailed in Table 1 and are in good agreement with Iwama et al. [24]: after 6 days, the process results in the deposition of slightly D-rich (−) ArgFum crystals and the mother liquor was strongly enriched with L(+)-ArgFum (i.e., ca. (+) 95%ee—green line in Figure 2). By contrast, the mother liquor should be at around (+) 70–80%ee and the crystals should be racemic or slightly enriched in L (see red line in Figure 2). If the process was performed with an opposite starting situation (i.e., with a slight excess of (−) 5%ee), the reverse final situation would be obtained: the mother liquor would be strongly enriched in D(−)-ArgFum and the crystals would be slightly enriched with L(+)-ArgFum. Concerning the deposited crystals, XRPD analyses by In-SituX^®^ confirmed the major presence of a single form whose structure has already been published (CSD refcode NOLPEL). The XRPD pattern of the pure anhydrous ArgFum enantiomer was never observed throughout this study.

Further to this, the incidence of other parameters was investigated by modification of the standard protocol. This involved stirring, process duration, initial supersaturation and starting ee.

Concerning stirring, if the use of magnetic stirring systematically drove the system to equilibrium (i.e., crystal in equilibrium with the mother liquor at ca. 70–80%ee—red equilibrium line in Figure 2) it was observed that the use of a rocking plate can improve slightly the speed of the process (see Appendix A) but may also stochastically induce failure. This illustrates the out-of-equilibrium nature of PE and that the process should be performed under stagnant conditions.To assess the kinetic stability of the final state of a typical PE experiment, the process duration was extended up to 1 year. The enantiomeric composition of the mother liquor and deposited crystal remained constant after several months and the onset of the return to equilibrium was only observed after 1 year (see Appendix A).The process was performed with initial supersaturation of β = 2, 4 and 6. Successful PE was observed at β = 4 and 6, but the experiments at β = 2 failed (see Appendix A), suggesting the existence of a supersaturation threshold below which PE cannot occur. Our experience suggests that the higher the supersaturation, the faster the process, although experiments with β > 8 were difficult to handle due to spontaneous crystallization before reaching T = 5 °C or directly in the mixing set-up. We did not investigate further such situations.The starting ee was varied from 0 to ca. (+) 20%ee. When the initial ee is exactly 0%, no one can predict the final sign of the mother liquor and the opposite sign of the solid. In the 0–1% range, the PE effect is weak. Above 1% there is a take-off in the final ee of the mother liquor which approaches 90% and then exceeds 95%. There is a threshold of the initial ee at ca. 11% above which the ee of the deposited crystals was of the same sign as that of the mother liquor.

### 2.2. Monitoring of the PE Phenomenon of Arginine Fumarate Using the Standard Protocol

To gain a deeper understanding of the mechanism, the mass balance of the process was established as a function of time. Chiral HPLC-UV was used to establish the evolution of the ee in the liquid and solid phases during the process. For the monitoring of the solid phase mass and ee, it was necessary to perform the standard protocol several times under identical conditions and to sacrifice it after different time lapses, from 0.25 to 144 h, after the beginning of crystallization. Once filtered, the whole solid fractions were weighed, then dissolved, diluted and analyzed by chiral HPLC-UV to measure the concentration of each enantiomer in the crystallized solid phase. Concerning the monitoring of the liquid phase, aliquots were sampled during the process and analyzed by chiral HPLC. The mass balance is presented in Figure 3a and the evolution of the ee is shown in Figure 3b,c. 

Figure 3a shows that 50% of the expected solid phase appeared within 15 min and that after 6 h of PE there was no more variation in the mass of the deposited crystals. The percentage of the expected solid phase is expressed with reference to the solubility of the DL-ArgFum phase at 5 °C. Figure 3b,c indicate that in the very early stage of the experiment, the ee of the deposited crystal was already opposite to the initial ee of the mother liquor: after 5 min, the first 50 mg of deposited crystals were indeed already at (−) 1%ee and the final solid phase ee was reached within an hour. This regime is actually faster than that observed in the case of the (DL)-phenylalanine fumaric acid salt reported by Gonnade in 2010 [27]. A similar regime is observed regarding the evolution of the ee of the liquid phase which attains its final level in a matter of hours. Actually, from 24 to 144 h, no measurable variation of the ee of the mother liquor was observed.

To further support these observations, it was decided to monitor the PE process using a different approach. After filtration, the collected crystals were subjected to a successive dissolution procedure which aimed at pilling off the different crystalline layers using the protocol described in the material and methods section. The washing fractions obtained during this procedure were analyzed by chiral HPLC-UV in order to establish the ee profile of the crystals, from the outer to the inner crystalline shells (i.e., corresponding to the later and earlier stages of crystallization, respectively). Such experiments permit rewinding the history of the crystals in terms of enantiomeric composition which is therefore an indirect way to monitor PE. Let us designate this method as “Monitoring by rewinding”. The results of these series of partial dissolution experiments are given in Figure 4 for different batches of crystals, filtered 2.5, 6, 24 and 144 h after the beginning of PE.

From Figure 4, one can see that the four profiles obtained are similar, with the external layers (the first 15%wt) having an ee of up to (+) 30% and the internal layers (the remaining 80%wt) a rather constant ee of ca. (−) 6%ee. These results are in good agreement with those obtained by full dissolution which showed an overall solid ee of about (−) 3 to (−) 4%ee (Table 1). The high L-ArgFum content of the outer shell being retrieved in the 4 profiles, should be regarded as an artifact, likely due to the presence of trapped L(+)-rich mother liquor released upon washing. The results obtained by this method confirm those obtained previously (Figure 3) and further demonstrate that neither the mass nor the enantiomeric composition of both the deposited solid and the mother liquor change after the first hours of monitoring.

With regards to the possible occurrence of a solvent-mediated polymorphic transformation in this system, further analyses were performed by monitoring the process by optical microscopy, in situ observations of the deposited crystals failed to reveal any solvent-mediated polymorphic transition and only revealed the presence of urchin-like crystal agglomerates during the whole process (optical microscopy pictures are shown in Appendix A and an SEM picture is shown in Figure 5). We did not observe the formation of the thin needle-shaped single crystals of the recently reported hydrated form of DL-ArgFum (CSD refcode ECANII) [26]. Some PE experiments were performed in the In-SituX^®^ reactor and were monitored in operando by X-ray diffraction without sampling [21,27]. The results of these experiments are shown as Appendix A. Despite the occurrence of unattributed diffraction peaks, most PE phenomena occurred while only the diffraction pattern of the original DL-ArgFum racemic form (CSD refcode NOLPEL) was detected (see Appendix A). The unattributed peaks are hardly imputable to the newly reported structure by Manoj et al. [26]. Several ex situ XRPD measurements were also performed at various times by sacrificing PE experiments, revealing the major presence of the stable DL-ArgFum “racemic” form.

Solvent-mediated polymorphic transitions have been regarded as an essential feature of the mechanisms of PE, at least for the first generation of compounds [16]. It acts as a trigger by which the first nucleating phase can “heal” its stacking fault upon redissolving the enantiomer in excess. It is a disruptive event involving nucleation that is discontinuous in nature. Our results did not reveal any hint of such transitions. The fact that PE starts as early as ArgFum crystallization begins and seems to operate in a continuous mode until the end of the process suggests a non-disruptive mechanism. Further studies were performed in order to comfort the hypothesis of a smooth process.

### 2.3. Doping the Process with ^13^C_6_-L-Arginine.HCl

The monitoring by rewinding method prompted us to tackle the mechanism of PE from a different perspective. Labeled L-Arginine (850 μg ^13^C_6_-L-Arginine.HCl in a 10 µL water solution, representing 0.1%wt of the total ArgFum mass) was voluntarily added in solution at different times (2.5, 6, 144 h) during PE. The standard process was performed either starting with a (+) or (−) 5%ee (if the process starts with an excess in (−)-D, the solution becomes strongly enriched in (−)-D and the deposited crystals are slightly enriched in (+)-L). Afterward, monitoring by rewinding was used to check if the L(+)-labeled-Arginine could be retrieved inside the deposited crystals, and if so, where. The results of the HPLC-MS analyses are given in Figure 6 and in Appendix A.

The comparison between the repartition profiles established for the crystals obtained with the two processes (i.e., starting with opposite ee: (+) or (–)) shows that almost no labeled L-arginine was detected in the deposited crystals if PE started with an excess in L (orange lozenges in Figure 6) whereas it was detected down to the 40%wt outmost crystal layers if PE started with an excess in D (other points in Figure 6). In the latter case, the repartition profiles are identical whatever the time of addition (i.e., 2.5 h, 6 h or 144 h), although it has been established that PE proceeds rather quickly and that no major mass transfer between the solid and liquid phases should occur ca. 1 h after the onset of crystallization.

## 3. Discussion

Figure 3 and Figure 4 show that PE starts as early as crystallization begins and is almost completed within ca. 1 h. Yet, the added L(+)-labeled arginine was shown to enter deeply into the crystal structure even if the addition was performed 144 h after PE ended (Figure 6). A direct consequence of this finding is that the occurrence of a solvent-mediated solid phase transition does not appear to take part in the PE mechanism of DL-ArgFum, in agreement with in situ X-ray diffraction measurements. In fact, our results cannot confirm the implication of the two recently reported forms of arginine fumarate hydrates in the PE process, as proposed by Manoj et al. [26]. One might even wonder if this is a necessary criterion for PE at all or if it is a fortuitous over-representation resulting from the Ostwald rule of stages [28] (under high supersaturation, the likelihood of crystallization of metastable phase is larger).

Furthermore, since labeled L(+)-arginine entered the L(+)-enriched crystals 144 h after PE completion, it can also be deduced: (i) PE is triggered by high initial supersaturation, (ii) once PE starts, exchanges between mother liquor and crystals occur regardless the degree of supersaturation. This striking process feature suggests that the mother liquor has the ability to expel its minor enantiomer by matter exchange with the crystals, which in turn permits it to reach maximum ee in the liquid. Although the mass balance is respected, this situation is in strong contradiction with the general rules of heterogeneous equilibria, since the tie lines cross the median line connecting the solvent and the racemic composition. Thus PE is a symmetry-breaking process that occurs far from equilibrium [29].

This leads us to reconsider the mechanism of PE, at least in the case of ArgFum. It seems that PE neither results from punctual events occurring during crystallization, nor from events such as solvent-mediated polymorphic transitions, but rather occurs as a consequence of two facts: (i) a continuous exchange of matter between the liquid and solid phase, and (ii) the much higher solubility of the pure enantiomer in the liquid phase compared to that of the racemic mixture. The essence of the driving force could therefore be related to the increased stability of ArgFum solutions with high ee, probably due to homochiral assemblies being more favorable than heterochiral ones. Although very high supersaturations are required in order to drive the system far from equilibrium and trap it in a metastable state, this parameter does not play a significative role in the mechanism of PE once this regime of exchange between the liquor and the solid phase has been reached. Figure 7 schematizes the mechanism of PE in light of the present findings.

An important consequence of this mechanism is that the solid phase formed during PE must tolerate some deviations from its racemic composition. Although In-SituX^®^ experiments revealed the occurrence of slight peak shifts occurring during the process (Appendix A), the ArgFum crystals obtained after 14 4h of PE were analyzed by SHG. This technique is sensitive to non-centrosymmetry in crystals and successfully evidenced the presence of enantio-enriched zones in ‘racemic’ crystals of histidine which exhibits a weak PE effect [30]. Figure 8a shows the presence of few and localized zones of non-centrosymmetry (in green) inside the ArgFum crystals (ca. few tenths of microns in size) obtained via the standard protocol, highlighting the presence of some enantio-enriched domains in the material (enlargements are provided as Appendix A). By comparison, racemic ArgFum crystals gently recrystallized from water hardly presented such zones (Figure 8b). If a homogeneous distribution of enantiomers inside the close-to-racemic crystals is likely to be unspotted (or hardly spotted) by SHG microscopy, the domains showing positive SHG signal are obviously due to much higher enantiomeric excesses. These enantio-enriched domains can be staked within a metastable solid solution matrix thanks to the match between lattice parameters. This could be put in parallel with the cases of “lamellar conglomerates” reported by the late Prof. Pivnitsky [31]. Figure 8a, therefore, shows that under the strong supersaturation conditions of PE, the crystal structure of DL-ArgFum accepts to deviate from its racemic composition, sometimes rather strongly, by exchanging enantiomers with the mother liquor, thus giving rise to such enantio-enriched zones.

A procedure of molecular modeling was applied to confirm that an enantiomeric permutation is energetically possible in the crystal structure of DL-ArgFum. It consists of sequentially switching the chirality of up to three arginine enantiomers inside a supercell (3 × 3 × 3) of the racemic structure of the DL-ArgFum salt. Thus, we artificially generate non-racemic solid solutions. The energetic cost of such enantiomeric substitutions was then computed by molecular modeling (semi-empirical level of theory) by comparing the lattice energies of the artificial structures to that of the racemic ones. The detailed procedure has been published elsewhere [23] and appears as a satisfactory tool to explain the existence (or the absence) of solid solutions between enantiomers [32]. The details of the modeling calculations performed for DL-ArgFum are given in Appendix A. It was found that, in the crystal structure of DL-ArgFum, these enantiomeric permutations have almost no detectable energetic cost (i.e., close to 0 kcal·mol^−1^). For comparison, switching an enantiomer in the structure of DL-Tryptophan Ethyl Ester Hydrochloride structure, a compound that does not show PE although exhibiting all the prerequisites mentioned in the Introduction, costs ca. 30 to 40 kcal·mol^−1^ [23]. These results are consistent with the possibility for the DL-ArgFum crystal structure to accommodate the deviation from the racemic composition imposed by the mother liquor.

## 4. Materials and Methods

### 4.1. Materials

DL-arginine 98% was purchased from Alfa Aesar (Schiltigheim, France), L-arginine 98%, fumaric acid 97% and D-Methionine 98% were purchased from Acros Organics (Geel, Belgium), ^13^C_6_-L-arginine.HCl 99% was purchased from Cambridge Isotope Laboratories (Andover, MA, USA). Methanol and ethanol were of HPLC grade and purchased from VWR (Fontenay-sous-Bois, France). Analytical grade formic acid was purchased from Sigma Aldrich (Saint-Quentin Fallavier, France). Water was ultra-purified (18.2 MΩ) with an Elga system from Veolia water solution & technologies (Saint-Maurice, France).

### 4.2. Recrystallization of DL-Arginine Fumarate

An amount of 4.35 g of fumaric acid and 6.5 g of DL-Arginine were dissolved in 35 mL of water at 80 °C. Crystallization of DL-arginine fumarate (DL-ArgFum) occurred upon stirring the solution at 5 °C overnight. An amount of 8.35 g (77% yield) of a white solid was obtained after filtration and drying. To reach precise enantiomeric compositions, adequate amounts of L-arginine and fumaric acid (L-ArgFum) were added and triturated with DL-ArgFum.

### 4.3. Procedure for PE Experiments

The following protocol was used to perform PE of DL-ArgFum by crystallization: In a 100 mL round-bottomed-flask, 850 mg of slightly enriched ArgFum (typically, (+)5%ee) was mixed with 1.7 mL of water. The suspension was stirred at atmospheric pressure by the motor system of a rotary evaporator at 80 °C, until complete dissolution. Then, using a syringe pump (0.5 mL/min), a volume of 1.7 mL of hot ethanol (60 °C) was gently added to the solution by slowly flowing the alcohol onto the inner wall of the flask. This quite unusual type of antisolvent addition permits a specific mixing regime between the two liquids allowing the solution to remain supersaturated. Uncontrolled crystallization occurs systematically during the addition if this operation is not carefully performed. Then the highly supersaturated solution was cooled down to 5 °C. After six days at this temperature without stirring, the obtained deposited crystals were filtered at 5 °C and both the mother liquor and the deposited crystals were analyzed by chiral HPLC-UV. In this paper, when the following conditions are used, the above protocol is referred to as “standard”: starting ArgFum mass m_ArgFum_ = 850 mg, supersaturation β = 8, crystallization temperature T_crist_ = 5 °C. The standard protocol was used as a reference for all the experiments performed in this study, some of them involving major modifications of the procedure (e.g., addition of labeled L-Arginine, stirring, supersaturation, etc…).

### 4.4. Procedure for Monitoring

To monitor the PE of DL-ArgFum, the standard protocol was performed in a large number of crystallization vials. At fixed times (i.e., 0.25, 0.50, 0.75,1, 1.5, 2.5, 6, 12, 24, 72 and 144 h), at least 3 experiments were stopped by filtration and the solid and liquid phases were analyzed using the HPLC-UV method. The ee of the liquid and solid phases, the amount of dissolved crystals in the mother liquor as well as the mass of deposited crystals were averaged over the 3 attempts.

### 4.5. Procedure for Successive Dissolution

It was necessary to analyze the compositions of the different layers of ArgFum deposited crystals. The following dissolution procedure was adopted to analyze the crystals obtained using the “Standard” PE Protocol (i.e., 850 mg scale). Freshly prepared ArgFum crystals were loaded into a jacketed sintered glass filter (n°5) with a temperature set at 5 °C. The glass filter was mounted on a regular vacuum filtration vessel. The crystals were (i) first quickly washed twice with 2 × 5 mL of a 90:10 EtOH/H2O mixture (T = 5 °C), this permits a washing of the remaining mother liquor (with e.e. >90% in L). Then (ii) 9 fractions of 5 mL (EtOH/H_2_O) 50:50 (T = 5 °C) were successively used, dissolving 15% of the overall crystal mass. The temperature of the glass filter was then increased to 45 °C and (iii) the crystals were washed with 9 fractions of 5mL pure water (T = 45 °C). These washings dissolved the remaining 85% of the crystal mass. It is worth noting that to promote dissolution during (ii) and (iii), the filtration vessel was agitated using a rocking plate (30 s, 15 rpm). On average, ca. 15 s was necessary to filter the different fractions. The different fractions were stored for further analysis. It was confirmed that the first fraction (i) had an ee higher than 90%.

### 4.6. Procedure for the Addition of Labeled L-Arginine

To investigate the mechanism of PE of DL-ArgFum, a minute amount of ^13^C_6_-L-Arginine representing 0.1%wt of the total mass of arginine fumarate (with reference to the standard protocol) was added to the mixture. This addition was made at different times after the start of the PE process (2.5, 6 or 144 h). After the addition, the system was left for 6 days at 5 °C before filtration and analyses.

### 4.7. HPLC-UV Method

The chromatographic analyses were performed using a liquid chromatograph from Thermo Fisher Scientific (Sunnyvale, CA, USA), an Ultimate 3000 equipped with an LPG-3400SD pump, an ACC-3000 autosampler, a 5 μL loop, a VWD-3400RS UV detector. Data processing was performed using Chromeleon 6.80 software (Thermo Fisher Scientific, Sunnyvale, CA, USA). The enantioselective separation was performed using a Supelco CHIROBIOTIC T column (150 mm × 2.1 mm × 5 µm) at 25 °C. The mobile phase was composed of 65% of methanol and 35% water containing 0.04% of formic acid. The flow rate was set at 0.2 mL/min. The detection wavelength was set at 207 nm. The injected sample volume was 5 µL.

### 4.8. HPLC-MS/MS Method

The chromatographic and mass spectrometry analyses were performed with an Agilent 1200 liquid chromatograph system (Agilent, Palo Alto, CA, USA) composed of a G1312A pump, a G1329A autosampler with a 5 μL loop, a G1316A oven and a G1314B UV detector, a Bruker HCT Ultra (Bruker Daltonics, Bremen, Germany) mass spectrometer, and HyStar 3.2 software (Bruker Daltonics, Bremen, Germany) for acquisition and data processing. The chromatographic separation was performed with the same conditions as the HPLC-UV method. Mass spectrometer conditions: ESI positive mode; Nebulizer 45 psi/300 °C; Dry gas 12 L/min Capillary 3000 V; Skimmer 40 V; CapExit 104.5 V; Amplitude fragmentation 0.29 (smart frag 30–200%)

### 4.9. Experimental Set-Up for In Situ XRPD Measurements

In situ XRPD data were collected using a homemade prototype diffractometer, In-SituX^®^ [33]. This apparatus has an original goniometer with an inverted geometry (−θ/−θ) associated with a dedicated reactor equipped with a transparent bottom for X-rays. It serves to identify solids in suspension during the crystallization process (time and temperature dependent), without any sampling. The detector is an LYNXEYE (Bruker, Germany) and the beam (Ni filtered) comes from an X-ray tube with a copper anticathode.

### 4.10. SHG Microscopy

An Insight X3 single laser with automated dispersion compensation (Spectra-Physics, Santa Clara, CA, USA) and a TCS SP8 MP confocal microscope (Leica Microsystems, Wetzlar, Germany) were used to perform confocal microscopy as well as two-photon microscopy and fluorescence lifetime imaging of the samples. The laser cavity had over 2.44 W of average power at 900 nm and was tunable from 680 nm to 1300 nm. The repetition rate was 80 MHz and the temporal width at the output of the cavity was around 120 fs (CS2) or Leica objective (HC PL APO 40× NA 1.30 CS2). An electro-optical modulator adjusted the laser power at the entrance of the confocal system. To check if the sample produces fluorescence, an emission spectral scan is performed. Typically, the sample is excited at a given wavelength (e.g., 1200 nm or 900 nm) while scanned through the emission wavelength (e.g., in the 385–780 nm range). The SHG signal appears at half of the excitation wavelength. The spectral acquisition was performed using an internal hybrid detector. Collected photons were dispersed by a prism and a specific motorized split mirror was used to select the spectral detection band before the hybrid detector. Acquisitions were performed between 385 nm and 780 nm every 3 nm with a spectral bandwidth of 5 nm. The samples were first sieved (50 < x < 100 µm) and prepared by deposition of a few mg on a microscope slide.

## 5. Conclusions

The present paper reports on the reinvestigation of the PE process of the arginine fumarate salt. Starting from a highly supersaturated racemic or slightly enriched solution (β = 8), the liquor becomes strongly enantio-enriched (>+90%ee) while the deposited crystals exhibit a weak opposite ee (i.e., <−5%ee). Our main findings bring new insights concerning the mechanism of PE of DL-ArgFum: (i) PE proceeds in less than 1 h after the onset of crystallization, (ii) PE proceeds via a continuous exchange of matter between the liquor and the crystals (iii) no polymorphic transition seems to be necessary for this process: solvent-mediated or not.

PE, rather than being the consequence of dramatic events such as polymorphic transitions, appears to be the consequence of peculiar interactions, involving evidence of exchange, between the mother liquor (which is stabilized by high ee), and the solid phase. The possibility for crystals to deviate from their racemic composition is also a key element of this process. The mechanism is likely to involve a metastable solid solution and/or the formation of enantio-enriched lamellae fitting well with the close-to-racemic crystals. Therefore, we conclude that the main pre-requisites for the occurrence of PE are (a) the solubility of the enantiomer is much higher than the solubility of the racemic mixture, (b) the occurrence of homochiral pre-associations in the solvated state, (c) an initial far from equilibrium crystallization, (d) absence of stirring/abrasive effect and (e) the existence of a solid solution close to the racemic composition and/or the formation of enantio-enriched domains without high energetic penalty.

## Data Availability

Not applicable.

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
