# Peer review of "Reinvestigating the Preferential Enrichment of DL-Arginine Fumarate: New Thoughts on the Mechanism of This Far from Equilibrium Crystallization Phenomenon"

_molecules, 2022, doi:10.3390/molecules27248652_

Round 1

Reviewer 1 Report

The MS by Coquerel et al. describes new approach to study preferential enantiomeric enrichment phenomenon during crystallization of Arginine Fumarate salt. The authors used isotopically labelled arginine as marker for chiral HPLC-MS. The MS is well-done and can be published in Molecules after taking these points into consideration:

1) What about the effect of solvent composition onto PE phenomenon? Did you try another water: ethanol mixtures? Vodka?

2) What about x-ray powder diffraction analysis of the crystalline material isolated in time? Authors mentioned some polymorphism but some additional details are needed for clarity.

3) The results of modelling are not clearly presented. Some more details should be added.

Reviewer 2 Report

The present paper reports on the reinvestigation of PE process of the Arginine Fumarate salt. It’s concluded that the main pre-requisites for the occurrence of PE are: (a) the solubility of the enantiomer is much higher than the solubility of the racemic mixture, (b) the occurrence of homochiral pre-associations in the solvated state, (c) an initial far from equilibrium crystallization, (d) absence of stirring/abrasive effect and (e) the existence of a solid solution close to the racemic composition without high energetic penalty. Overall, the work is comprehensive while extensive experimental as well as model effort has been made to present new evidence on the specific field of research.

Recommendation: major revision.

 Comments:

1. Lines 29 and 30, “A chiral substance can exist as two stereoisomers called enantiomers”. It should be more precise, because not all chiral materials have only one chiral center.

2. The crystal structure of racemic salts should be analyzed specific more, whether there is a pure enantiomer domain or whether the crystal structure contains disorder. Is there any evidence that a crystal is a solid solution?

3. There may be a conflict between Line 104, “no other polymorph of this substance has been reported” and lines 116-118.

4. Lines 164-166, “Our experience suggests that the higher the supersaturation, the faster the process, although experiments with β>8 were difficult to handle due to spontaneous crystallization before reaching T=5°C.” I want to know the results of spontaneous crystallization with β>8. Wouldn't it be spontaneous crystallization under other conditions β=4,6? Are there any seeds?

5. What basis is the Mass percentage of deposited ArgFum crystals as a function of time in Figure 3 calculated?  What expected mass is? Mass percentage of deposited ArgFum crystals as a function of time?

6. In fact, PE is crystallized many times, but each time the crystal will crystallize into a basically fixed ee. Is there any thermodynamic basis? Why the stable nonracemic mixed crystals capable of memorizing the event of chiral symmetry breaking?

7. In ref [24], it have already reached the conclusion that direct nucleationfrom the molecular clusters present in the supersaturated solution to give the stable mixed crystalline phase may be inducing the chiral symmetry breaking and thereby PE (rather than a polymorphic transition). Please summarize your new innovations outside of this work.

8. The range of the solid solution seems to be a small region close to the racemic composition. Does the boundary of the region matter? Could PE operate on a full range of solid solution system?

9. Can PE operate on an epitaxial conglomerate system?

10. The association of homophile chains in solution seems to be very important in PE. Is there any evidence to support the true occurrence of homophile association in solution?

11. For an experiment with an initial condition of 0%ee, is there any relation between the uncertainty of the direction of resolution and the uncertainty of the deracemization direction of viedma ripening?

12. What drives symmetry breaking far away from equilibrium?

13. As previously reported, the initial crystal formation was a metastable solid solution, which was then dissolved driven by phase transition, resulting in an excess of enantiomers. But in this work, the nuclei that are initially formed are actually racemic with association of homophile chains in solution, which is confusing.

14. After 144h of experiment, isotopically labeled L solution was added, which could still penetrate deep into the crystal, that is to say, solid-liquid exchange was still going on, and it was L growth and D dissolution. But in fact, after 6 hours, the ee of the solution doesn't change anymore. Is it because the addition of L solution affects the ee of the system? Why not add isotopically labeled DL racemic solution?

Author Response

see file attached.
